# Efficient and Flexible Inference for Stochastic Systems

**Stefan Bauer**[*]
Department of Computer Science
ETH Zurich
bauers@inf.ethz.ch

**Nico S. Gorbach**[*]
Department of Computer Science
ETH Zurich
ngorbach@inf.ethz.ch

**Đorđe Miladinović**
Department of Computer Science
ETH Zurich
djordjem@inf.ethz.ch

**Joachim M. Buhmann**
Department of Computer Science
ETH Zurich
jbuhmann@inf.ethz.ch

## Abstract

Many real world dynamical systems are described by stochastic differential equations. Thus parameter inference is a challenging and important problem in many disciplines. We provide a grid free and flexible algorithm offering parameter and state inference for stochastic systems and compare our approch based on variational approximations to state of the art methods showing significant advantages both in runtime and accuracy.

## 1 Introduction

A dynamical system is represented by a set of $K$ stochastic differential equations (SDE's) with model parameters $\boldsymbol{\theta}$ that describe the evolution of $K$ states $\mathbf{X}(t) = [x_1(t), x_2(t), \ldots, x_K(t)]^T$ such that:

$$d\mathbf{X}(t) = \mathbf{f}(\mathbf{X}(t), \boldsymbol{\theta})dt + \boldsymbol{\Sigma}d\mathbf{W}_t, \tag{1}$$

where $\mathbf{W}_t$ is a Wiener process. A sequence of observations, $\mathbf{y}(t)$ is usually contaminated by some measurement error which we assume to be normally distributed with zero mean and variance for each of the $K$ states, i.e. $\mathbf{E} \sim \mathcal{N}(\mathbf{0}, \mathbf{D})$, with $\mathbf{D}_{ik} = \sigma_k^2 \delta_{ik}$. Thus for $N$ distinct time points the overall system may be summarized as

$$\mathbf{Y} = \mathbf{AX} + \mathbf{E},$$

where

$$\mathbf{X} = [\mathbf{x}(t_1), \ldots, \mathbf{x}(t_N)] = [\mathbf{x}_1, \ldots, \mathbf{x}_K]^T$$
$$\mathbf{Y} = [\mathbf{y}(t_1), \ldots, \mathbf{y}(t_N)] = [\mathbf{y}_1, \ldots, \mathbf{y}_K]^T,$$

where $\mathbf{x}_k = [x_k(t_1), \ldots, x_k(t_N)]^T$ is the k'th state sequence and $\mathbf{y}_k = [y_k(t_1), \ldots, y_k(t_N)]^T$ are the observations. Given the observations $\mathbf{Y}$ and the description of the dynamical system (1), the aim is to estimate both state variables $\mathbf{X}$ and parameters $\boldsymbol{\theta}$.

**Related Work.** Classic approaches for solving the inverse problem i.e. estimating the parameters given some noisy observations of the process, include the Kalman Filter or its improvements [e.g. Evensen, 2003, Tornøe et al., 2005] and MCMC based approaches [e.g. Lyons et al., 2012]. However,

---

[*]The first two authors contributed equally to this work.

MCMC based methods do not scale well since the number of particles required for a given accuracy grows exponentially with the dimensionality of the inference problem [Snyder et al., 2008], which is why approximations to the inference problem became increasingly more popular in recent years. Archambeau et al. [2008] proposed a variational formulation for parameter and state inference of stochastic diffuion processes using a linear dynamic approximation: In an iterated two-step approach the mean and covariance of the approximate process (forward propagation) and in the second step the time evolution of the Lagrange multipliers, which ensure the consistency constraints for mean and variance (backward propagation), are calculated in order to obtain a smooth estimate of the states. Both forward and backward smoothing require the repeated solving of ODEs. In order to obtain a good accuracy a fine time grid is additionally needed, which makes the approach computational expensive and infeasible for larger systems [Vrettas et al., 2015]. For parameter estimation the smoothing algorithm is used in the inner loop of a conjugate gradient algorithm to obtain an estimate of the optimal approximation process (given a fixed set of parameters) while in the outer loop a gradient step is taken to improve the current estimate of the parameters. An extension of Archambeau et al. [2008] using local polynomial approximations and mean-field approximations was proposed in Vrettas et al. [2015]. Mean-field approximations remove the need of Lagrange multipliers and thus of the backward propagation while the polynomial approximations remove the need of solving ODEs iteratively in the forward propagation step which makes the smoothing algorithm and thus the inner loop for parameter estimation feasible, even for large systems while achieving a comparable accuracy [Vrettas et al., 2015].

**Our contributions.** While established methods often assume full observability of the stochastic system for parameter estimation, we solve the more difficult problem of inferring parameters in systems which include unobserved variables by combining state and parameter estimation in one step. Despite the fact that we compare our approach to other methods which solve a simpler problem, we offer improved accuracy in parameter estimation at a fraction of the computational cost.

## 2 Random Ordinary Differential Equations

Compared to stochastic differential equations, random ordinary differential equations (RODEs) have been less popular even though both frameworks are highly connected. RODEs are pathwise ordinary differential equations that contain a stochastic process in their vector field functions. In Kloeden and Jentzen [2007] RODEs have been studied to derive better numerical integration schemes for SDEs, which e.g. allows for stronger pathwise results compared to the $L_2$ results given in Ito stochastic calculus. Moreover, RODEs sometimes have an advantage over SDEs by allowing more realistic noise for some applications e.g. correlated noise or noise with limited variance. Let $(\Omega, \mathcal{F}, \mathcal{P})$ be a complete probability space, $(\zeta_t)_{t \in [0,T]}$ be a $\mathcal{R}^m$-valued stochastic process with continuous sample paths and $f : \mathcal{R}^m \times \mathcal{R}^d \to \mathcal{R}^d$ a continuous function. Then

$$\frac{d\mathbf{x}(t)}{dt} = \mathbf{f}(\mathbf{x}(t), \zeta_t(\omega)) \tag{2}$$

is a scalar RODE, that is, an ODE

$$\frac{d\mathbf{x}(t)}{dt} = F_\omega(t, x) := \mathbf{f}(\mathbf{x}(t), \omega(t)), \tag{3}$$

for all $\omega \in \Omega$. Following Kloeden and Jentzen [2007], we likewise assume that $f$ is arbitrary smooth i.e. $f \in \mathcal{C}^\infty$ and thus locally Lipschitz in $x$ such that the initial value problem (3) has a unique solution, which we assume to exist on the finite time interval $[0, T]$. A simple example for a RODE is

**Example 1** (RODE)**.**

$$\frac{d\mathbf{x}(t)}{dt} = -x + \sin(W_t(\omega)), \tag{4}$$

where $W_t$ is a Wiener process. Taylor-like schemes for directly solving RODEs (2) were derived e.g. in Grüne and Kloeden [2001], Jentzen and Kloeden [2009]. One approach for solving the RODE (2) is to use sampling to obtain many ODE's (3) which can then be solved pathwise using deterministic calculus. However, this pathwise solution of RODEs implies that a massive amount of deterministic ODEs have to be solved efficiently. A study with a high performance focus was conducted in

Riesinger et al. [2016], where parallelized pathwise inference for RODEs was implemented using GPU's. While in principle classic numerical schemes for deterministic systems e.g. Runge-Kutta can be used for each path, they will usually converge with a lower order since the vector field is not smooth enough in time [Asai et al., 2013]. Since the driving stochastic process $\zeta_t$ has at most Hölder continuous sample paths, the sample paths of the solution $t \to x(t)$ are continuously differentiable but the derivatives of the solution sample paths are at most Hölder continuous in time. This is caused by the fact that $F_\omega(t, x)$ of the ODE (3) is usually only continuous, but not differentiable in $t$, no matter how smooth the function $f$ is in its variables. RODEs offer the opportunity to use deterministic calculus (pathwise), yet being highly connected with an SDE since any RODE with a Wiener process can be written as SDE Jentzen and Kloeden [2011]. To illustrate the point, the example 1 above can be re-written as an SDE by:

**Example 2** (SDE transformed RODE).

$$d \begin{pmatrix} X_t \\ Y_t \end{pmatrix} = \begin{pmatrix} -X_t + \sin(Y_t) \\ 0 \end{pmatrix} + \begin{pmatrix} 0 \\ 1 \end{pmatrix} dW_t. \tag{5}$$

It likewise holds that SDEs can be transformed into RODEs. This transformation was first described in Sussmann [1978] and Doss [1977] and generalized to all finite dimensional stochastic differential equations by Imkeller and Schmalfuss [2001]. RODEs can thus be used to find pathwise solutions for SDEs but SDEs can likewise be used to find better solution for RODEs Asai and Kloeden [2013].

Due to space limitations and to circumvent the introduction of a large mathematical framework, we only show the transformation for additive SDE's following [Jentzen and Kloeden, 2011, chapter 2].

**Proposition 1.** *Any finite dimensional SDE can be transformed into an RODE and the other way round:*

$$d\mathbf{x}_t = \mathbf{f}(\mathbf{x}_t)dt + d\mathbf{W}_t \Longleftrightarrow \frac{d\mathbf{z}(t)}{dt} = \mathbf{f}(\mathbf{z}_t + \mathbf{O}_t) + \mathbf{O}_t, \tag{6}$$

*where* $\mathbf{z}(t) := \mathbf{x}_t - \mathbf{O}_t$ *and* $\mathbf{O}_t$ *is the Ornstein-Uhlenbeck stochastic stationary process satisfying the linear SDE*

$$d\mathbf{O}_t = -\mathbf{O}_t dt + d\mathbf{W}_t \tag{7}$$

Typically a stationary Ornstein-Uhlenbeck process is used to replace the white noise of the SDE in its transformation to an RODE. By continuity and the Fundamental Theorem of Calculus it then follows that $\mathbf{z}(t)$ is pathwise differentiable. While we only showed the transformation for additive SDE's, it generally holds true that any RODE with a Wiener process can be transformed into an SDE and any finite dimensional SDE with regular coefficients can be transformed into an RODE. This includes nonlinear drifts and diffusions and is true for univariate and multivariate processes [Han and Kloeden, 2017]. There are cases for which this does not hold e.g. a RODE which includes fractional Brownian motion as the driving noise. While the presented method is thus even more general since RODE's can be solved, we limit ourselves to the problem of solving additive SDE's by transforming them into a RODE.

Since the solution of a RODE is continuously differentiable in time (but not further differentiable in time), classic numerical methods for ODEs rarely do achieve their traditional order and thus efficiency [Kloeden and Jentzen, 2007]. In the following we describe a scalable variational formulation to infer states and parameters of stochastic differential equations by providing an ensemble learning type algorithm for inferring the parameters of the corresponding random ordinary differential equation.

## 3    Variational Gradient Matching

Gradient matching with Gaussian processes was originally motivated in Calderhead et al. [2008] and offers a computationally efficient shortcut for parameter inference in deterministic systems. While the original formulation was based on sampling, Gorbach et al. [2017] proposed a variational formulation offering significant runtime and accuracy improvements.

Gradient matching assumes that the covariance kernel $\mathbf{C}_{\phi_k}$ (with hyper-parameters $\phi_k$) of a Gaussian process prior on state variables is once differentiable to obtain a conditional distribution over state

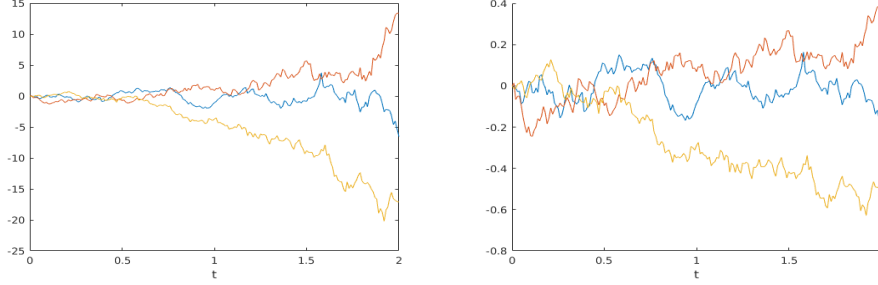

Figure 1: **Noise.** The left plot shows three typical Wiener processes generated with mean zero and the corresponding Ornstein-Uhlenbeck (OU) process having the same Wiener process in its diffusion (right). The scale on the y-axis shows the mean-reverting behaviour of the OU process (compared to the Wiener process).

derivatives using the closure property under differentiation of Gaussian processes:

$$p(\dot{\mathbf{X}} \mid \mathbf{X}, \boldsymbol{\phi}) = \prod_k \mathcal{N}(\dot{\mathbf{x}}_k \mid \mathbf{m}_k, \mathbf{A}_k), \tag{8}$$

where the mean and covariance is given by:

$$\mathbf{m}_k := {}'\mathbf{C}_{\boldsymbol{\phi}_k} \mathbf{C}_{\boldsymbol{\phi}_k}^{-1} \mathbf{x}_k, \quad \mathbf{A}_k := \mathbf{C}_{\boldsymbol{\phi}_k}'' - {}'\mathbf{C}_{\boldsymbol{\phi}_k} \mathbf{C}_{\boldsymbol{\phi}_k}^{-1} \mathbf{C}_{\boldsymbol{\phi}_k}',$$

$\mathbf{C}_{\boldsymbol{\phi}_k}''$ denotes the auto-covariance for each state-derivative with $\mathbf{C}_{\boldsymbol{\phi}_k}'$ and $'\mathbf{C}_{\boldsymbol{\phi}_k}$ denoting the cross-covariances between the state and its derivative.

The posterior distribution over state-variables is

$$p(\mathbf{X} \mid \mathbf{Y}, \boldsymbol{\phi}, \boldsymbol{\sigma}) = \prod_k \mathcal{N}\left(\boldsymbol{\mu}_k(\mathbf{y}_k), \boldsymbol{\Sigma}_k\right), \tag{9}$$

where $\boldsymbol{\mu}_k(\mathbf{y}_k) := \mathbf{C}_{\boldsymbol{\phi}_k}(\mathbf{C}_{\boldsymbol{\phi}_k} + \sigma_k^2 \mathbf{I})^{-1} \mathbf{y}_k$ and $\boldsymbol{\Sigma}_k := \sigma_k^2 \mathbf{C}_{\boldsymbol{\phi}_k}(\mathbf{C}_{\boldsymbol{\phi}_k} + \sigma_k^2 \mathbf{I})^{-1}$.

Inserting the GP based prior in the right hand side of a differential equation and assuming additive, normally distributed noise with state-specific error variance $\gamma_k$ one obtains a distribution of state derivatives

$$p(\dot{\mathbf{X}} \mid \mathbf{X}, \boldsymbol{\theta}, \boldsymbol{\gamma}) = \prod_k \mathcal{N}\left(\dot{\mathbf{x}}_k \mid \mathbf{f}_k(\mathbf{X}, \boldsymbol{\theta}), \gamma_k \mathbf{I}\right). \tag{10}$$

which is combined with the smoothed distribution obtained from the data fit (9) in a product of experts approach:

$$p(\dot{\mathbf{X}} \mid \mathbf{X}, \boldsymbol{\theta}, \boldsymbol{\phi}, \boldsymbol{\gamma}) \propto p(\dot{\mathbf{X}} \mid \mathbf{X}, \boldsymbol{\phi}) p(\dot{\mathbf{X}} \mid \mathbf{X}, \boldsymbol{\theta}, \boldsymbol{\gamma}).$$

After analytically integrating out the latent state-derivatives

$$p(\boldsymbol{\theta} \mid \mathbf{X}, \boldsymbol{\phi}, \boldsymbol{\gamma}) \propto p(\boldsymbol{\theta}) \prod_k \mathcal{N}\left(\mathbf{f}_k(\mathbf{X}, \boldsymbol{\theta}) \mid \mathbf{m}_k, \boldsymbol{\Lambda}_k^{-1}\right). \tag{11}$$

where $\boldsymbol{\Lambda}_k^{-1} := \mathbf{A}_k + \gamma_k \mathbf{I}$ one aims to determine the maximum a posteriori estimate (MAP) of the parameters

$$\boldsymbol{\theta}^* := \arg\max_{\boldsymbol{\theta}} \ln \int p(\boldsymbol{\theta} \mid \mathbf{X}, \boldsymbol{\phi}, \boldsymbol{\gamma}) p(\mathbf{X} \mid \mathbf{Y}, \boldsymbol{\phi}) d\mathbf{X}, \tag{12}$$

Since the integral in (12) is in most cases analytically intractable (even for small systems due to the non-linearities and couplings induced by the drift function), a lower bound is established through the

introduction of an auxiliary distribution $Q$:

$$\ln \int p(\boldsymbol{\theta} \mid \mathbf{X}, \boldsymbol{\phi}, \boldsymbol{\gamma})p(\mathbf{X} \mid \mathbf{Y}, \boldsymbol{\phi})d\mathbf{X}$$

$$\stackrel{(a)}{=} - \int Q(\mathbf{X})d\mathbf{X} \, \ln \frac{\int Q(\mathbf{X})d\mathbf{X}}{\int p(\boldsymbol{\theta} \mid \mathbf{X}, \boldsymbol{\phi}, \boldsymbol{\gamma})p(\mathbf{X} \mid \mathbf{Y}, \boldsymbol{\phi})d\mathbf{X}}$$

$$\stackrel{(b)}{\geq} - \int Q(\mathbf{X}) \ln \frac{Q(\mathbf{X})}{p(\boldsymbol{\theta} \mid \mathbf{X}, \boldsymbol{\phi}, \boldsymbol{\gamma})p(\mathbf{X} \mid \mathbf{Y}, \boldsymbol{\phi})}d\mathbf{X}$$

$$= H(Q) + \mathbb{E}_Q \ln p(\boldsymbol{\theta} \mid \mathbf{X}, \boldsymbol{\phi}, \boldsymbol{\gamma}) + \mathbb{E}_Q \ln p(\mathbf{X} \mid \mathbf{Y}, \boldsymbol{\phi})$$

$$=: \mathcal{L}_Q(\boldsymbol{\theta}) \tag{13}$$

where $H(Q)$ is the entropy. In (a) the auxiliary distribution $Q(\mathbf{X})$, $\int Q(\mathbf{X})d\mathbf{X} = 1$ is introduced and in (b) is using Jensens's inequality. The lower bound holds with equality whenever

$$Q^*(\mathbf{X}) := \frac{p(\boldsymbol{\theta} \mid \mathbf{X}, \boldsymbol{\phi}, \boldsymbol{\gamma})p(\mathbf{X} \mid \mathbf{Y}, \boldsymbol{\phi})}{\int p(\boldsymbol{\theta} \mid \mathbf{X}, \boldsymbol{\phi}, \boldsymbol{\gamma})p(\mathbf{X} \mid \mathbf{Y}, \boldsymbol{\phi})d\mathbf{X}} \stackrel{(c)}{=} p(\mathbf{X} \mid \mathbf{Y}, \boldsymbol{\theta}, \boldsymbol{\phi}, \boldsymbol{\gamma}),$$

where in (c) Bayes rule is used. Unfortunately $Q^*$ is analytically intractable because its normalization given by the integral in the denominator is in most cases analytically intractable due to the strong couplings induced by the nonlinear drift function $f$ in (1). Using mean-field approximations

$$\mathcal{Q} := \left\{ Q : Q(\mathbf{X}, \boldsymbol{\theta}) = q(\boldsymbol{\theta} \mid \boldsymbol{\lambda}) \prod_u q(\mathbf{x}_u \mid \boldsymbol{\psi}_u) \right\}, \tag{14}$$

where $\boldsymbol{\lambda}$ and $\boldsymbol{\psi}_u$ are the variational parameters. Assuming that the drift in (1) is linear in the parameters $\boldsymbol{\theta}$ and that states only appear as monomial factors in arbitrary large products of states the true conditionals $p(\boldsymbol{\theta} \mid \mathbf{X}, \mathbf{Y}, \boldsymbol{\phi})$ and $p(\mathbf{x}_u \mid \boldsymbol{\theta}, \mathbf{X}_{-u}, \mathbf{Y}, \boldsymbol{\phi})$ are Gaussian distributed, where $\mathbf{X}_{-u}$ denotes all states excluding state $\mathbf{x}_u$ (i.e. $\mathbf{X}_{-u} := \{\mathbf{x} \in \mathbf{X} \mid \mathbf{x} \neq \mathbf{x}_u\}$) and thus $q(\boldsymbol{\theta} \mid \boldsymbol{\lambda})$ and $q(\mathbf{x}_u \mid \boldsymbol{\psi}_u)$ are designed to be Gaussian.

This posterior distribution over states is then approximated as $p(\mathbf{X} \mid \mathbf{Y}, \boldsymbol{\theta}, \boldsymbol{\phi}, \boldsymbol{\gamma}, \boldsymbol{\sigma}) \approx \widehat{Q}(\mathbf{X}) = \prod_k \prod_t \widehat{q}_{\boldsymbol{\psi}_{kt}}$ and the log transformed distribution over the ODE parameters given the observations as $\ln p(\boldsymbol{\theta} \mid \mathbf{Y}, \boldsymbol{\phi}, \boldsymbol{\gamma}, \boldsymbol{\sigma}) \approx \mathcal{L}_{\widehat{Q}}(\boldsymbol{\theta})$.

---

**Algorithm 1** Ensemble based parameter estimation for SDEs

---

1: Transform the SDE 1 into a RODE 2
2: Simulate a maximum number $N_{\max}$ of OU-processes and insert them in 2 to obtain $N_{\max}$ ODEs
3: For each ODE obtain approximate solutions using variational gradient matching [Gorbach et al., 2017]
4: Combine the solutions $\widehat{\boldsymbol{\theta}}$ to obtain an estimate of the parameters for the RODE 2
5: Transform the solutions of the RODE 2 back into solutions of the SDE 1.

---

Gorbach et al. [2017] then use an EM-type approach illustrated in figure 2 iteratively optimizing parameters and the variational lower bound $\mathcal{L}_{\widehat{Q}}(\boldsymbol{\theta})$. The variational parameters can be derived analytically and the algorithm scales linearly in the number states of the differential equation and is thus ideally suited to infer the solutions of the massive number of pathwise ODEs required for the pathwise solution of the RODE formulation of the SDE. Since solution paths of the RODE are only once differentiable, gradient matching (which only makes this assumption w.r.t. solution paths) is ideally suited for estimating the parameters. Our approach is summarized in algorithm 1.

However, the application of variational gradient matching [Gorbach et al., 2017] for the pathwise solution of the RODE is not straightforward since e.g. in the case for scalar stochastic differential equations one has to solve

$$\frac{d\mathbf{z}(t)}{dt} = f_\theta(z_t + O_t) + O_t, \tag{15}$$

for a sampled trajectory $O_t$ of an Ornstein-Uhlenbeck process rather than the classic ODE formulation $\frac{d\mathbf{z}(t)}{dt} = f(z_t)$. We account for the increased uncertainty by assuming an additional state specific Gaussian noise factor $\delta$ i.e. assuming $f(x + O_t) + O_t + \delta$ for a sampled trajectory $O_t$ in the gradient matching formulation (10).

**Flexibility and Efficiency** Algorithm 1 offers a flexible framework for inference in stochastic dynamical systems e.g. if the parameters $\hat{\theta}$ are known they can be set to the true values in each iteration, and algorithm 1 then just corresponds to a smoothing algorithm. Compared to the smoothing algorithm in Archambeau et al. [2008] it does not require the computational expensive forward and backward propagation using an ODE solver. If the parameters are not known then algorithm 1 offers a grid free inference procedure for estimating the parameters. Opposite to Vrettas et al. [2011] which consider unobserved state variables in the case of smoothing but assume the system to be fully observed if parameters are estimated, the outlined approach offers an efficient inference framework for the much more complicated problem of inferring the parameters while not all states are observed and still scales linearly in the states if pathwise inference of the RODE is done in parallel.

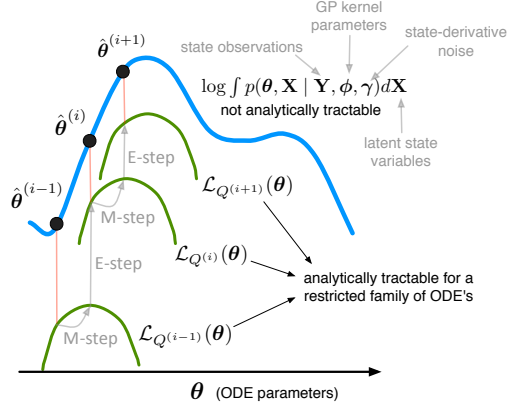

Figure 2: Illustration of the "hill climbing" algorithm in Gorbach et al. [2017] . The difference between the lower bound $\mathcal{L}_{\widehat{Q}^{(\cdot)}}(\theta)$ and the log integral is given by the Kullback-Leibler divergence (red line).

The conceptual difference between the approach of Vrettas et al. [2015] and Gorbach et al. [2017] is illustrated in figure 3.

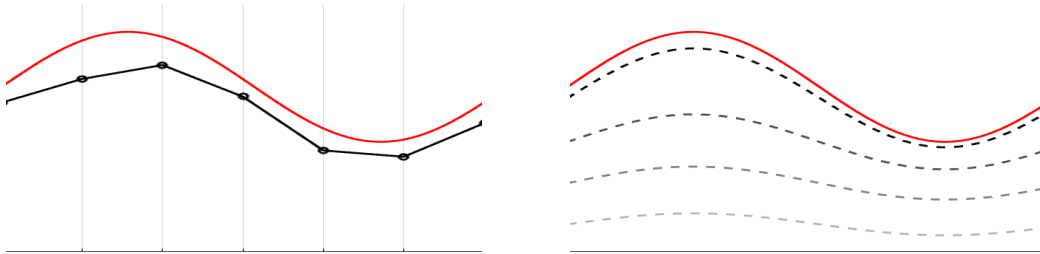

Figure 3: **Conceptual Difference.** The red line represents an artificial function which has to be approximated. Our approach (right) is grid free and based on the minimization of the differences of the slopes. That is why convergence is vertical with each iteration step corresponding to a dashed line (thickness of the line indicating the convergence direction). Vrettas et al. [2015] approximate the true process by a linearized dynamic process which is discretized (left) and improved by iterated forward and backward smoothing.

# 4    Experiments

We compare our approach on two established benchmark models for stochastic systems especially used for weather forecasts. Vrettas et al. [2011] provide an extensive comparison of the approach of Archambeau et al. [2008] and its improvements compared to classic Kalman filtering as well as more advanced and state of the art inference schemes like 4D-Var [Le Dimet and Talagrand, 1986]. We use the reported results there as a comparison measure.

The drift function for the Lorenz96 system consists of equations of the form:

$$f_k(\mathbf{x}(t), \boldsymbol{\theta}) = (x_{k+1} - x_{k-2})x_{k-1} - x_k + \theta$$

where $\theta$ is a scalar forcing parameter, $x_{-1} = x_{K-1}, x_0 = x_K$ and $x_{K+1} = x_1$ (with $K$ being the number of states in the stochastic system (1)). The Lorenz96 system can be seen as a minimalistic weather model [Lorenz and Emanuel, 1998].

The three dimensional Lorenz attractor is described by the parameter vector $\boldsymbol{\theta} = (\sigma, \rho, \beta)$ and the following time evolution:

$$d\mathbf{X}(t) = \begin{bmatrix} \sigma(x_2(t) - x_1(t)) \\ \rho x_1(t) - x_2(t) - x_1(t)x_3(t) \\ x_1(t)x_2(t) - \beta x_3(t) \end{bmatrix} dt + \boldsymbol{\Sigma}^{\frac{1}{2}} d\mathbf{W}_t$$

The runtime for state estimation using the approach of Vrettas et al. [2011] and our method is indicated in table 1. While parameter and state estimation are combined in one step in our approach, parameter estimation using the approach of Vrettas et al. [2011] would imply the iterative use of the smoothing algorithm and thus a multiple factor of the runtime indicated in table 1. While we solve a much more difficult problem by inferring parameters and states at the same time our runtime is only a fraction of the runtime awarded for a single run of the inner loop for parameter estimation in Vrettas et al. [2011].

| Method | L63/D=3 | L96/D=40 | L96/D=1000 |
|---|---|---|---|
| VGPA_MF | 31s | 6503s | 17345s |
| Our approach | 2.4s | 14s | 383s |

Table 1: Runtime for one run of the smoothing algorithm of the approach of Vrettas et al. [2015] vs the runtime of our approach in parallel implementation (using 51 OU sample paths). While parameter estimation is done simultaneously in our approach, Vrettas et al. [2015] use the smoothing algorithm iteratively for state estimation in an inner loop such that the runtime for parameter estimations is multiple times higher than the indicated runtime for just one run of the smoothing algorithm.

We use our method to infer the states and drift parameters for the Lorenz attractor where the dimension $y$ is unobserved. The estimated state trajectories are shown in figure 4.

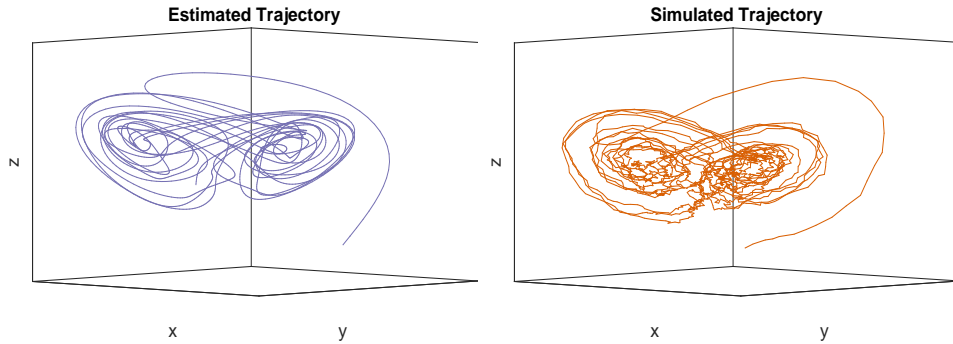

Figure 4: **Lorenz attractor.** The Lorenz attractor trajectories are shown on the right -hand side for inferred solutions using an SDE solver, while the left-hand side plot shows the inferred trajectory using our method. Our method was able to accurately resolve the typical "butterfly" pattern despite not observing the drift parameters as well as not observing the dimension $y$. Only the dimensions $x$ and $z$ were observed.

The estimated trajectories for one sample path are also shown in the time domain in section 5.2 of the supplementary material.

Our approach offers an appealing shortcut to the inference problem for stochastic dynamical systems and is robust to the noise in the diffusion term. Figure 5 shows the dependence of the inferred parameters on the variance in the diffusion term of the stochastic differential equation.

Increasing the time interval of the observed process e.g. from 10 to 60 secs leads to a converging behaviour to the true parameters (figure 6). This is in contrast to the reported results of Archambeau et al. [2008], reported in Vrettas et al. [2011, Figure 29] and shows the asymptotic time consistency of our approach.

Figure 5 shows, that in the near noiseless scenario we approximately identify sigma correctly. Estimating the $\sigma$ term in Figure 6 is more difficult than the other two parameters in the drift

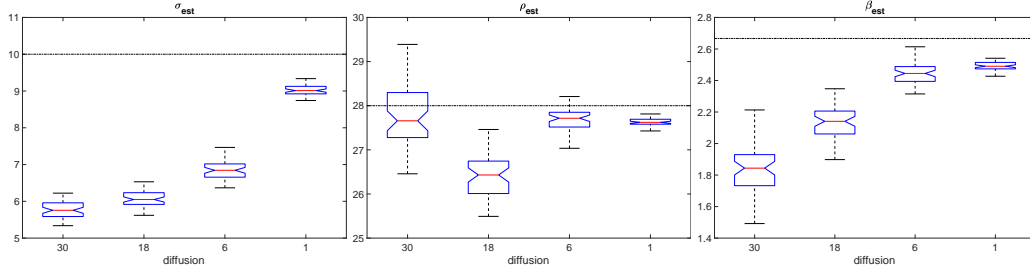

Figure 5: **Lorenz attractor.** Boxplots indicate the median of the inferred parameters over 51 generated OU sample paths. Using a low variance for the diffusion term in simulating *one* random sample path from the SDE, our approach infers approximately the correct parameters and does not completely deteriorate if the variance is increased by a factor of 30.

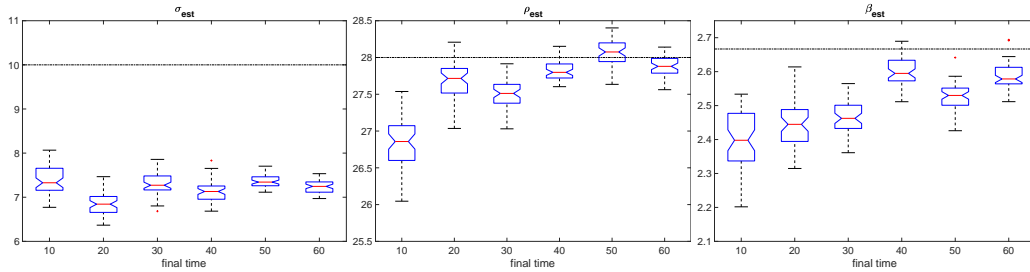

Figure 6: **Lorenz attractor.** Increasing the time interval of the observed process leads to a convergence towards the true parameters opposed to the results in [Vrettas et al., 2011, Figure 29].

function of the Lorenz attractor system, since the variance of the diffusion and the observation noise unfortunately lead to an identifiability problem for the parameter sigma, which is why longer time periods in Figure 6 do not improve the estimation accuracy for $\sigma$.

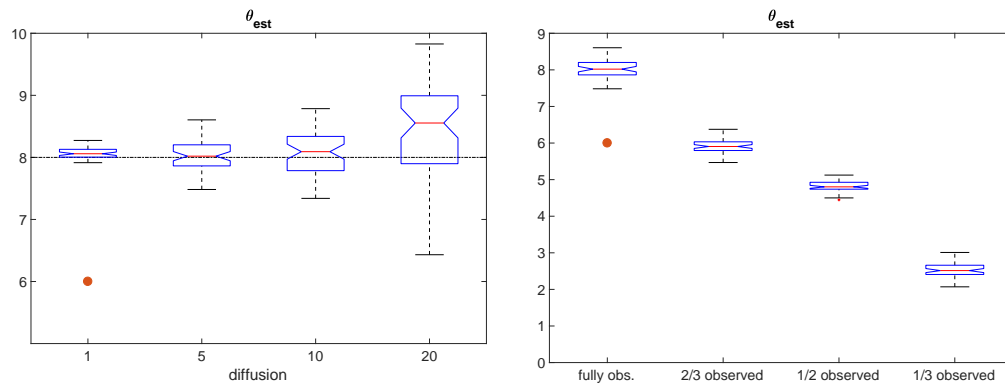

Figure 7: **Lorenz96.** Left hand side shows the accuracy of the parameter estimation with increasing diffusion variance (right to left) for a 40 dimensional system, while the plot on the right hand side shows the accuracy with decreasing number of observations. Red dots show the results of the approach of Archambeau et al. [2008] when available as reported in Vrettas et al. [2011]. The correct parameter has the value 8 and our approach performs significantly better, while having a lower runtime and is furthermore able to include unobserved variables (right)

For the Lorenz96 system our parameter estimation approach is likewise robust to the variance in the diffusion term (figure 7). It furthermore outperforms the approach of Archambeau et al. [2008] in the cases where results were reported in Vrettas et al. [2011]. The performance level is equal when, for our approach, we assume that only *one third* of the variables are unobserved.

The estimated trajectories for one sample path of the Lorenz96 system are shown in section 5.3 of the supplementary material.

## 5 Discussion

Parameter inference in stochastic systems is a challenging but important problem in many disciplines. Current approaches are based on exploration in the parameter space which is computationally expensive and infeasible for larger systems. Using a gradient matching formulation and adapting it to the inference of random ordinary differential equations, our proposal is a flexible framework which allows to use deterministic calculus for inference in stochastic systems. While our approach tackles a much more difficult problem by combining state and parameter estimation in one step, it offers improved accuracy and is orders of magnitude faster compared to current state of the art methods based on variational inference.

## Acknowledgements

This research was partially supported by the Max Planck ETH Center for Learning Systems and the SystemsX.ch project SignalX.

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
