[Supplementary Material]

# Supplementary Material for:
## "Efficient and Flexible Inference for Stochastic Systems"

### 5.1 Update equations for Parameter and State Estimation

Due to the model assumption in Gorbach et al. [2017], the true conditional distributions $p(\boldsymbol{\theta} \mid \mathbf{X}, \mathbf{Y}, \boldsymbol{\phi}, \boldsymbol{\gamma}, \boldsymbol{\sigma})$ and $p(\mathbf{x}_u \mid \boldsymbol{\theta}, \mathbf{X}_{-u}, \mathbf{Y}, \boldsymbol{\phi}, \boldsymbol{\gamma}, \boldsymbol{\sigma})$ are Gaussian distributed, where $\mathbf{X}_{-u}$ denotes all states excluding state $\mathbf{x}_u$ (i.e. $\mathbf{X}_{-u} := \{\mathbf{x} \in \mathbf{X} \mid \mathbf{x} \neq \mathbf{x}_u\}$). For didactical reasons, we write the true conditional distributions in canonical form:

$$p(\boldsymbol{\theta} \mid \mathbf{X}, \mathbf{Y}, \boldsymbol{\phi}) = h(\boldsymbol{\theta}) \times \exp\left(\boldsymbol{\eta}_{\boldsymbol{\theta}}(\mathbf{X}, \mathbf{Y}, \boldsymbol{\phi}, \boldsymbol{\gamma}, \boldsymbol{\sigma})^T \mathbf{t}(\boldsymbol{\theta}) - a_{\boldsymbol{\theta}}(\boldsymbol{\eta}_{\boldsymbol{\theta}}(\mathbf{X}, \mathbf{Y}, \boldsymbol{\phi}, \boldsymbol{\gamma}, \boldsymbol{\sigma}))\right)$$

$$p(\mathbf{x}_u \mid \boldsymbol{\theta}, \mathbf{X}_{-u}, \mathbf{Y}, \boldsymbol{\phi}) = h(\mathbf{x}_u) \times \exp\left(\boldsymbol{\eta}_u(\boldsymbol{\theta}, \mathbf{X}_{-u}, \mathbf{Y}, \boldsymbol{\phi})^T \mathbf{t}(\mathbf{x}_u) - a_u(\boldsymbol{\eta}_u(\mathbf{X}_{-u}, \mathbf{Y}, \boldsymbol{\phi}))\right)$$

The particular form of $q(\boldsymbol{\theta} \mid \boldsymbol{\lambda})$ and $q(\mathbf{x}_u \mid \boldsymbol{\psi}_u)$ of the mean-field approximations 14, are designed to be in the same exponential family as the true conditional distributions:

$$q(\boldsymbol{\theta} \mid \boldsymbol{\lambda}) := h(\boldsymbol{\theta}) \exp\left(\boldsymbol{\lambda}^T \mathbf{t}(\boldsymbol{\theta}) - a_{\boldsymbol{\theta}}(\boldsymbol{\lambda})\right)$$

$$q(\mathbf{x}_u \mid \boldsymbol{\psi}_u) := h(\mathbf{x}_u) \exp\left(\boldsymbol{\psi}_u^T \mathbf{t}(\boldsymbol{\psi}_u) - a_u(\boldsymbol{\psi}_u)\right),$$

whose optimal variational parameters are given by:

$$\hat{\boldsymbol{\lambda}} := \begin{pmatrix} \mathbb{E}_Q \boldsymbol{\Omega}_{\boldsymbol{\theta}}^{-1} \mathbf{r}_{\boldsymbol{\theta}} \\ -\frac{1}{2} \mathbb{E}_Q \boldsymbol{\Omega}_{\boldsymbol{\theta}}^{-1} \end{pmatrix}, \tag{16}$$

$$\hat{\boldsymbol{\psi}}_u := \begin{pmatrix} \mathbb{E}_Q \boldsymbol{\Omega}_u^{-1} \mathbf{r}_u \\ -\frac{1}{2} \mathbb{E}_Q \boldsymbol{\Omega}_u^{-1} \end{pmatrix} \tag{17}$$

Next we analytically derive the parameters $\mathbf{r}_{\boldsymbol{\theta}}$ and $\boldsymbol{\Omega}_{\boldsymbol{\theta}}$ in equation 16 which we obtain from the true conditional distribution over ODE parameters:

$$
\begin{aligned}
p(\boldsymbol{\theta} \mid \mathbf{X}, \mathbf{Y}, \boldsymbol{\phi}, \boldsymbol{\gamma}) &\stackrel{(a)}{=} p(\boldsymbol{\theta} \mid \mathbf{X}, \boldsymbol{\phi}, \boldsymbol{\gamma}) \\
&\stackrel{(b)}{=} Z_{\boldsymbol{\theta}}^{-1}(\mathbf{X}) \int p(\dot{\mathbf{X}} \mid \mathbf{X}, \boldsymbol{\theta}, \boldsymbol{\phi}, \boldsymbol{\gamma}) p(\dot{\mathbf{X}} \mid \mathbf{X}, \boldsymbol{\phi}) d\dot{\mathbf{X}} \\
&\stackrel{(c)}{=} Z_{\boldsymbol{\theta}}^{-1}(\mathbf{X}) \prod_k \mathcal{N}\left(\mathbf{f}_k(\mathbf{X}, \boldsymbol{\theta}) \mid \mathbf{m}_k, \boldsymbol{\Lambda}_k^{-1}\right) \\
&\stackrel{(d)}{=} Z_{\boldsymbol{\theta}}^{-1}(\mathbf{X}) \prod_k \mathcal{N}\left(\mathbf{B}_{\boldsymbol{\theta}k} \boldsymbol{\theta} + \mathbf{b}_{\boldsymbol{\theta}k} \mid \mathbf{m}_k, \boldsymbol{\Lambda}_k^{-1}\right) \\
&\stackrel{(e)}{=} Z_{\boldsymbol{\theta}}'^{-1}(\mathbf{X}) \prod_k \mathcal{N}\left(\boldsymbol{\theta} \mid \left(\mathbf{B}_{\boldsymbol{\theta}k}^T \boldsymbol{\Lambda}_k \mathbf{B}_{\boldsymbol{\theta}k}\right)^{-1} \mathbf{B}_{\boldsymbol{\theta}k}^T \boldsymbol{\Lambda}_k (\mathbf{m}_k - \mathbf{f}_{\boldsymbol{\theta}k}), \left(\mathbf{B}_{\boldsymbol{\theta}k}^T \boldsymbol{\Lambda}_k \mathbf{B}_{\boldsymbol{\theta}k}\right)^{-1}\right) \\
&\stackrel{(f)}{=} \mathcal{N}\left(\boldsymbol{\theta} \mid \mathbf{r}_{\boldsymbol{\theta}}, \boldsymbol{\Omega}_{\boldsymbol{\theta}}\right),
\end{aligned}
$$

where $Z_{\boldsymbol{\theta}}(\mathbf{X})$ and $Z_{\boldsymbol{\theta}}'(\mathbf{X})$ normalize the distributions and $\mathbf{m}_k$ and $\boldsymbol{\Lambda}_k$ are defined as above. In (a) we notice that $\boldsymbol{\theta}$ does not directly depend on the observations $\mathbf{Y}$ but instead indirectly through the states $\mathbf{X}$. In (b) we substitute the product of experts and in (c) we analytically integrate out the state derivatives. We rewrite the ODE $\mathbf{f}_k$ as a linear combination of the ODE parameters (i.e. $\mathbf{B}_{\boldsymbol{\theta}k} \boldsymbol{\theta} + \mathbf{b}_{\boldsymbol{\theta}k} \stackrel{!}{=} \mathbf{f}_k(\mathbf{X}, \boldsymbol{\theta})$) in (d) and in (e) we normalize each factor w.r.t. $\boldsymbol{\theta}$. In (g) we normalize the product of Gaussians where mean and covariance are given by:

$$\mathbf{r}_{\boldsymbol{\theta}} := \boldsymbol{\Omega}_{\boldsymbol{\theta}} \sum_k \mathbf{B}_{\boldsymbol{\theta}k}^T \boldsymbol{\Lambda}_k (\mathbf{m}_k - \mathbf{b}_{\boldsymbol{\theta}k}), \qquad \boldsymbol{\Omega}_{\boldsymbol{\theta}}^{-1} := \sum_k \mathbf{B}_{\boldsymbol{\theta}k}^T \boldsymbol{\Lambda}_k \mathbf{B}_{\boldsymbol{\theta}k}.$$

Similarly, we analytically derive the parameters $\mathbf{r}_u$ and $\boldsymbol{\Omega}_u$ in equation 17 which we obtain from the true conditional distribution over an individual state:

$$p(\mathbf{x}_u \mid \boldsymbol{\theta}, \mathbf{X}_{/\{\mathbf{x}_u\}}, \mathbf{Y}, \boldsymbol{\phi}, \boldsymbol{\gamma}) = Z_u^{-1}(\boldsymbol{\theta}) \prod_k \mathcal{N}\left(\mathbf{f}_k(\mathbf{X}, \boldsymbol{\theta}) \mid \mathbf{m}_k, \boldsymbol{\Lambda}_k^{-1}\right) \mathcal{N}\left(\mathbf{x}_u \mid \boldsymbol{\mu}_u(\mathbf{Y}), \boldsymbol{\Sigma}_u\right)$$

$$\stackrel{(g)}{=} Z_u^{-1}(\boldsymbol{\theta}) \prod_k \mathcal{N}\left(\mathbf{B}_{uk}\mathbf{x}_u + \mathbf{b}_{uk} \mid \mathbf{m}_k, \boldsymbol{\Lambda}_k^{-1}\right) \mathcal{N}\left(\mathbf{x}_u \mid \boldsymbol{\mu}_u(\mathbf{Y}), \boldsymbol{\Sigma}_u\right)$$

$$= Z_u'^{-1}(\boldsymbol{\theta}) \prod_k \mathcal{N}\left(\mathbf{x}_u \mid \left(\mathbf{B}_{uk}^T \boldsymbol{\Lambda}_k \mathbf{B}_{uk}\right)^{-1} \mathbf{B}_{uk}^T \boldsymbol{\Lambda}_k (\mathbf{m}_k - \mathbf{b}_{uk}), \left(\mathbf{B}_{uk}^T \boldsymbol{\Lambda}_k \mathbf{B}_{uk}\right)^{-1}\right)$$

$$\mathcal{N}\left(\mathbf{x}_u \mid \boldsymbol{\mu}_u(\mathbf{Y}), \boldsymbol{\Sigma}_u\right)$$

$$= \mathcal{N}\left(\mathbf{x}_u \mid \mathbf{r}_u, \boldsymbol{\Omega}_u\right),$$

where $Z_u(\boldsymbol{\theta})$ and $Z_u'(\boldsymbol{\theta})$ normalize the distributions. For a partially observed system, the mean $\boldsymbol{\mu}_u(\mathbf{Y})$ and covariance $\boldsymbol{\Sigma}_u$ are given by $\boldsymbol{\mu}_u(\mathbf{Y}) := \sigma^{-2}\left(\sigma^{-2}\mathbf{A}^T\mathbf{A} + \mathbf{C}_\phi\right)^{-1}\mathbf{A}^T\mathbf{Y}$ and $\boldsymbol{\Sigma}_u^{-1} := \sigma^{-2}\mathbf{A}^T\mathbf{A} + \mathbf{C}_\phi^{-1}$, with matrix $\mathbf{A}$ accommodating for unobserved states by encoding the linear relationship between latent states and observations (i.e. $\mathbf{Y} = \mathbf{A}\mathbf{X} + \mathbf{E}$, $\mathbf{E} \sim \mathcal{N}\left(\mathbf{0}, \sigma^2\mathbf{I}\right)$). Once more, in (e) we define $\mathbf{B}_{uk}$ and $\mathbf{b}_{uk}$ such that the ODE $\mathbf{f}_k$ is expressed as a linear combination of the state $\mathbf{x}_k$ (i.e. $\mathbf{B}_{uk}\mathbf{x}_u + \mathbf{b}_{uk} \stackrel{!}{=} \mathbf{f}_k(\mathbf{X}, \boldsymbol{\theta})$). The mean and covariance are given by:

$$\mathbf{r}_u := \boldsymbol{\Omega}\left(\sum_k \mathbf{B}_{uk}^T \boldsymbol{\Lambda}_k (\mathbf{m}_k - \mathbf{b}_{uk}) + \boldsymbol{\Sigma}_u^{-1}\boldsymbol{\mu}_u(\mathbf{Y})\right), \qquad \boldsymbol{\Omega}_u^{-1} := \sum_k \mathbf{B}_{uk}^T \boldsymbol{\Lambda}_k \mathbf{B}_{uk} + \boldsymbol{\Sigma}_u^{-1}.$$

## 5.2 Lorenz Attractor

Parameter and state estimation has previously been shown in the experiments section 4 where we show the phase plots in figure 4. In figure 8 we show the same estimation in the time domain using mean-field gradient matching by Gorbach et al. [2017] for one sample path.

Figure 8: Mean-field gradient matching [Gorbach et al., 2017] is used for parameter and state estimation for the partially observed Lorenz attractor SDE with the $y$ dimension (bottom left) remaining unobserved. Red trajectories respectively denote the sample paths (some of which are observed) and purple denotes their estimation. Both the drift parameters as well as the states and most notably the unobserved state $y$ are estimated well. The experimental setup is the same as the one used by Vrettas et al. [2011]. We included another sample path (green) to illustrate the stochasticity of the system.

## 5.3 Lorenz 96

Parameter and state estimation for the Lorenz 96 SDE has also been shown previously in the experiments section 4. In figure 9 we additionally show the parameter and state trajectory estimation using mean-field gradient matching by Gorbach et al. [2017] for one sample path.

Figure 9: Mean-field gradient matching is used for parameter and state estimation for the partially observed Lorenz 96 SDE. The first $8$ of $40$ dimensions are shown. Red trajectories respectively denote the sample paths (some of which are observed) and purple denotes their estimation. Both the drift parameters as well as the unobserved state trajectories are estimated well. The experimental setup is the same as the one used by Vrettas et al. [2011]. We included another sample path (green) to illustrate the stochasticity of the system.