[Reviews · NeurIPS 2017]

Reviewer 1



***Update following reviewer discussion and author feedback*** I am happy to revise my score for the paper provided the authors add some discussion of the number of OU processes used in the simulations, the \delta term in the RODE system and add the volatility to the Lorenz96 model (in addition to the other changes to the text recommended by myself and the other reviewers). ************************************************************** The authors propose a method for combined state and parameter estimation for stochastic differential equation (SDE) models. The SDE is first transformed into a random ordinary differential equation. Several solution paths are then simulated to generate a large number of ordinary differential equations, and each of these is then solved using an EM algorithm type approach that was introduced in an earlier paper. The method is tested on two systems, the Lorenz96 and Lorenz63 models, and compared to a competitor method showing that the new approach can be faster and more accurate. There are some interesting ideas in the paper but I can’t accept it for publication in its current form. The general approach seems reasonable, but there are some details of it that the authors don’t really mention that I think need to be explored. There are also lots of things that I found unclear in the manuscript and I think these need to be fixed. Since there is no major revision option for the conference I must therefore recommend rejection. I think if the authors revise the paper accordingly it could be accepted at a similar conference in the future. Detailed comments: I think is section 3 you need to spend more time discussing two things: How many OU processes you should be simulating. You don’t seem to discuss it but it must be crucial to the performance of the approach in practice. Why the additional \delta term is added to each RODE system, and how its variance should be chosen. The main reason for these suggestions is that the approach itself borrows innovations from other sources (e.g. Gorbich et al. 2017). Because of this, the specifics of how these tools are applied in the SDE context is to my mind the main contribution of the present paper. If you don’t discuss these things then it isn’t clear how useful the paper is. The first line states ‘A dynamical system is represented by a set of K stochastic differential equations…’ This is not true in general, many dynamical systems cannot be represented by stochastic differential equation models (e.g. when the driving noise is not Brownian). line 56 why is equation (2) a scalar RODE? I would have thought that it was a d-dimensional system of RODEs. Or does scalar not mean what I think it means here? In (2) the vector field is described as f(x(t),w(t)), whereas in (2) it is written f(x(t),\eta(w)). Shouldn’t it instead be f(x(t),\eta(t)) in (2) and f(x(t),\eta(t,w)) in (3), since \eta(t) is a r.v. and \eta(t,w) will be the fixed outcome for w \in \Omega? Line 60. Here you introduce Example 1, but then fail to discuss it for the next paragraph. Then you introduce Example 2 which is essentially the second part of example 1. I think it would be better if you combined example 1 and 2, since the purpose of the example is to show how a RODE can be re-written as an SDE when the driving noise is Brownian. And also put it where example 2 currently is (i.e. below the paragraph of text). Line 87. ‘Stationary' for the OU process can either mean it is started at equilibrium or that it is positive recurrent (i.e. it will be ergodic). Which of these do you mean here? From the definition I think it should be the latter since the SDE could be conditioned to start at any point, but this could certainly cause some confusion I think. Line 89. Why an OU process? Can the authors offer any intuition here, as it seems to come out of nowhere and I think the reader would be very interested to understand the origins. Line 167. Is the Lorenz96 system therefore an SDE with unit volatility? I think it would be good to clarify this for non-experts. Typos line 24 diffusions processes -> diffusion processes line 44 the fact the we -> the fact that we line 48 ‘both frameworks are highly related’ doesn’t make sense. If two things are related then the statement clearly applies to both of them. line 54 is a R^m valued -> be an R^m valued line 99 an computationally efficient -> a computationally efficient line 113 an product of experts -> a product of experts line 119 an lower bound -> a lower bound line 174 in pur approach -> in our approach (I think?) line 190 robuts -> robust line 198 random ordinary differential equation -> random ordinary differential equations

Reviewer 2



*** Update following Author's feedback *** As my main concern was clarity and this issue is well addressed by the authors, I am raising the score to reflect my assessment of the potential impact of this paper. *** This paper provides a novel inference algorithm for estimating the parameters of stochastic differential equations (SDEs) with noisy observations, as well as inferring the posterior distribution over states. The proposed algorithm transforms the SDE into a random ordinary differential equation (RODE) and samples a set of ODEs with noisy observations from the RODE (with a minor modification of adding additional noise). Next, it applies a recently developed variational inference procedure for ODEs with noisy observations [Gorbach etal. 2017]. Finally, it transforms the RODE back to SDE. This approach is scalable in the dimension of the hidden state and it is tested on a 3, 40 and 1000 dimensional systems. The main contribution is the idea to leverage inference algorithms for ODE with noisy observations to solve SDE using RODE. This approach may open the way to develop additional algorithms for SDEs. The overall structure and the ideas in the paper are clearly written, but the math of the variational gradient matching is hard to follow. Importantly, update equations for Q are missing, though the reader is referred to another paper. Additionally, the modification of the gradient matching algorithm involves adding a noise factor delta (line 146). It would be helpful to specify how is this parameter determined. As the algorithm has a sampling component, a discussion about how does the number of OU-processes N_max affect the accuracy of the algorithm will be very informative. Some details are missing in the experimental section: - It is not clear from the text how does hyperparameters such as phi and the state specific error gamma are set in practice. - In the runtime comparison (Table 1), do the different methods achieve comparable accuracy? - In Figure 6 we see convergence of rho and beta towards the true parameter. Sigma seems not to be affected by the time interval. - To which version of the L96 system does Figure 7 refer to? (what is D) Minor comments and typos: - Figure 2 is too small as well as the fonts in other figures - Line 174: change pur to our - Line 182: the word ‘Figure’ is missing between system and 4. - Line 190: change robuts to robust - Line 192: ‘is equal’ is written twice